# TNF licenses macrophages to undergo rapid caspase-1, -11, and -8-mediated cell death that restricts *Legionella pneumophila* infection

**Tzvi Y. Pollock[1], Víctor R. Vázquez Marrero[1], Igor E. Brodsky[2], Sunny Shin** [1] *

**1** Department of Microbiology, University of Pennsylvania Perelman School of Medicine, Philadelphia, Pennsylvania, United States of America, **2** Department of Pathobiology, University of Pennsylvania School of Veterinary Medicine, Philadelphia, Pennsylvania, United States of America

* sunshin@pennmedicine.upenn.edu

**Data Availability Statement:** All relevant data are within the manuscript and its Supporting information files.

## Abstract

The inflammatory cytokine tumor necrosis factor (TNF) is necessary for host defense against many intracellular pathogens, including *Legionella pneumophila*. *Legionella* causes the severe pneumonia Legionnaires' disease and predominantly affects individuals with a suppressed immune system, including those receiving therapeutic TNF blockade to treat autoinflammatory disorders. TNF induces pro-inflammatory gene expression, cellular proliferation, and survival signals in certain contexts, but can also trigger programmed cell death in others. It remains unclear, however, which of the pleiotropic functions of TNF mediate control of intracellular bacterial pathogens like *Legionella*. In this study, we demonstrate that TNF signaling licenses macrophages to die rapidly in response to *Legionella* infection. We find that TNF-licensed cells undergo rapid gasdermin-dependent, pyroptotic death downstream of inflammasome activation. We also find that TNF signaling upregulates components of the inflammasome response, and that the caspase-11-mediated non-canonical inflammasome is the first inflammasome to be activated, with caspase-1 and caspase-8 mediating delayed pyroptotic death. We find that all three caspases are collectively required for optimal TNF-mediated restriction of bacterial replication in macrophages. Furthermore, caspase-8 is required for control of pulmonary *Legionella* infection. These findings reveal a TNF-dependent mechanism in macrophages for activating rapid cell death that is collectively mediated by caspases-1, -8, and -11 and subsequent restriction of *Legionella* infection.

## Author summary

The immune signaling molecule tumor necrosis factor (TNF) is critical for defense against many bacterial infections. In the absence of TNF, opportunistic bacterial pathogens such as *Legionella pneumophila*, which causes the severe pneumonia Legionnaires' disease, readily invade and replicate inside innate immune cells known as macrophages. While it is clear that TNF signaling is required for protection of macrophages against pathogens like *L. pneumophila*, the mechanisms by which this signaling protects cells from infection

**Funding:** This work was supported by National Institutes of Health (NIH)/National Institute of Allergy and Infectious Diseases (NIAID) grants: AI118861 (S.S.), AI123243 (S.S.), and AI151476 (S.S. and T.Y.P.); AI128530 (I.E.B.), AI135421 (I.E.B), and AI139102 (I.E.B.); AI140508 (T.Y.P.) and AI141393 (T.Y.P.); the Linda Pechenik Montague Investigator Award from the University of Pennsylvania Perelman School of Medicine (S.S.), the Burroughs-Wellcome Fund Investigators in the Pathogenesis of Infectious Disease Award (S.S. and I.E.B.), and the National Science Foundation Graduate Research Fellowship DGE-1650114 (V.R.V.M). The funders had no role in study design, data collection and analysis, decision to publish, or preparation of the manuscript.

**Competing interests:** The authors declare that no competing interests exist.

remain unclear. Here, we show that one of the methods by which TNF protects macrophages from *L. pneumophila* infection is by promoting the death of infected cells. TNF signaling can lead to the production of anti-microbial factors and inflammation, but it can also prompt cells to engage programmed forms of cell death. We demonstrate that in *L. pneumophila* infection, TNF specifically licenses cells to more rapidly and robustly activate a kind of cell death called pyroptosis, which causes inflammation and warns nearby uninfected cells. We additionally demonstrate several cooperative pathways downstream of TNF by which this death occurs. Our work shows that TNF protects macrophage populations from *L. pneumophila* by licensing cells to rapidly self-sacrifice upon infection and limit the replicative niche of the bacteria.

## Introduction

The innate immune system is generally capable of preventing and controlling infection. However, pathogens are able to establish infection when key immune factors are evaded or deficient [1]. Inflammatory cytokine signaling constitutes one such category of immune factors. The loss or inhibition of cytokine signaling often results in greatly increased risk of infection. Tumor Necrosis Factor (TNF) is one such critical mediator of host defense against intracellular pathogens [2–6]. Therapeutic blockade of TNF in the context of autoinflammatory diseases results in greatly increased rates of infection [7,8]. While the downstream effects of TNF signaling are well-established in regulating cell survival, proliferation, and death in many sterile contexts, it remains unclear how TNF controls intracellular pathogens.

Among the bacteria which require TNF for efficient immune clearance is the gram-negative, facultative intracellular bacterial pathogen *Legionella pneumophila*. The etiologic agent of the severe pneumonia Legionnaire's disease, *L. pneumophila* is an opportunistic pathogen that causes disease predominantly in immunocompromised hosts [9,10]. *L. pneumophila* replicates within host macrophages by using a Dot/Icm type IV secretion system (T4SS) to translocate more than 300 bacterial effectors [11–15]. T4SS effectors modulate numerous host processes, including membrane trafficking and protein translation [16–19]. *L. pneumophila* infection is typically well-controlled by the innate immune system [20,21]. This immune protection, however, is severely attenuated in the absence of TNF signaling in macrophages and mice or in autoimmune patients receiving TNF blockade [7,8,20,22–28].

A key aspect of the innate immune response to *L. pneumophila* involves inflammasomes, multi-protein cytosolic complexes which assemble in response to pathogenic insult and mediate downstream inflammatory signaling [20,29–32]. Delivery of *Legionella* flagellin into the host cytosol triggers the NAIP5/NLRC4 inflammasome in mice, activating the cysteine protease caspase-1 to induce an inflammatory form of cell death known as pyroptosis, processing and release of interleukin-1 family cytokines, and restriction of *L. pneumophila* replication [33–35]. *L. pneumophila* T4SS activity further activates the NLRP3 inflammasome, while cytosolic detection of *L. pneumophila* LPS activates caspase-11 to mediate the "non-canonical" inflammasome [36,37]. Inflammation and cell death mediated by inflammasomes restrict bacterial infection, in part by limiting the replicative niche of the pathogen and promoting cytokine production by bystander immune cells [38–40]. TNF is required for optimal NAIP5/NLRC4-mediated restriction of *L. pneumophila* replication within macrophages. However, TNF can also mediate restriction of *L. pneumophila* in the absence of NAIP5/NLRC4 inflammasome activation via a proposed cell death-independent mechanism that invokes the activity of caspases other than caspases-1 and -11 [27]. Thus, it is unclear whether TNF-mediated

control of *L. pneumophila* replication observed in the absence of the NAIP5/NLRC4 inflammasome is due to caspase-mediated cell death or other cellular fates.

TNF signaling in infectious and non-infectious settings can lead to multiple outcomes. Upon binding of TNF to its cognate receptors, TNF Receptor 1 (TNFR1) or TNF Receptor 2 (TNFR2), the adaptor protein TRADD is recruited to the cytosolic domain of the receptor to allow formation of functionally distinct signaling complexes [3,4,41]. Complex I mediates immune activation, inflammation, and production of anti-apoptotic survival factors through downstream NF-κB and MAPK signaling [4,42]. In contrast, Complex II can engage two distinct forms of programmed cell death: extrinsic apoptosis mediated by the cysteine protease caspase-8 cleaving and activating the executioner caspases-3 and -7 under situations when NF-κB and MAPK signaling are inhibited [4,5,43], or necroptosis mediated by receptor-interacting protein kinase 3 (RIPK3) when caspase-8 activity is blocked or absent [44–46]. Additionally, caspase-8 promotes inflammatory gene expression independently of its cell death function [43,47–49], as well as promotes caspase-1 activation and compensates for its absence in inflammasome activation to process gasdermin-D and IL-1 family cytokines [50–54].

In this study, we sought to define the mediators of bacterial restriction that function independently of the NAIP/NLRC4 inflammasome in the context of *L. pneumophila* infection. Our data demonstrate that TNF signaling through TNFR1 licenses macrophages to undergo more rapid and robust cell death in response to *L. pneumophila* infection. We found that this cell death was associated with gasdermin-dependent loss of membrane integrity, indicating that this cell death is pyroptotic. Our findings indicate that TNF licenses cells to upregulate and rapidly activate the non-canonical caspase-11 inflammasome in response to *L. pneumophila* infection. We further found that caspase-1 and caspase-8 contributed to delayed cell death in the absence of caspase-11. In addition, we found that caspase-8 contributed to cell death independently of its ability to execute extrinsic apoptosis. Moreover, caspase-8 activity was required for clearance of pulmonary *L. pneumophila* infection. These data together indicate that TNF signaling during *L. pneumophila* infection restricts bacterial replication by licensing macrophages to rapidly undergo caspase-1, -8, and -11 inflammasome activation and pyroptosis, thereby eliminating the replicative niche of the bacteria.

## Results

### TNFR1 signaling is required for restriction of *L. pneumophila* infection and licenses cells to rapidly undergo cell death

Macrophages infected with *L. pneumophila* rapidly respond to injected bacterial flagellin via the NAIP5/NLRC4 inflammasome, initiating pyroptotic cell death and eliminating the replicative niche of the bacteria [33,55]. Maximal NAIP5/NLRC4-dependent restriction requires TNF signaling [20]. However, TNF also restricts intracellular replication of *L. pneumophila* within macrophages independently of flagellin and the NAIP inflammasome [27], implying that TNF-dependent control of *L. pneumophila* likely involves multiple downstream innate responses. For the entirety of this study, we infected bone marrow-derived macrophages (BMDMs) with mutant *L. pneumophila* deficient for flagellin (Δ*flaA*) in order to bypass the NAIP5 inflammasome response and specifically examine the NAIP5/NLRC4-independent role of TNF signaling in control of infection. Consistent with previous studies, we observed that BMDMs isolated from wild-type (WT) C57BL/6 mice harbor between 10-fold and 100-fold replication of flagellin-deficient bacteria over the course of 72 hours, while BMDMs from *Tnf*$^{-/-}$ mice demonstrated significantly higher levels of bacteria at 48 and 72 hours postinfection (Fig 1A). Of note, TNF-dependent restriction was apparent only after 24 hours postinfection, indicating that endogenous TNF produced in response to the initial infection was

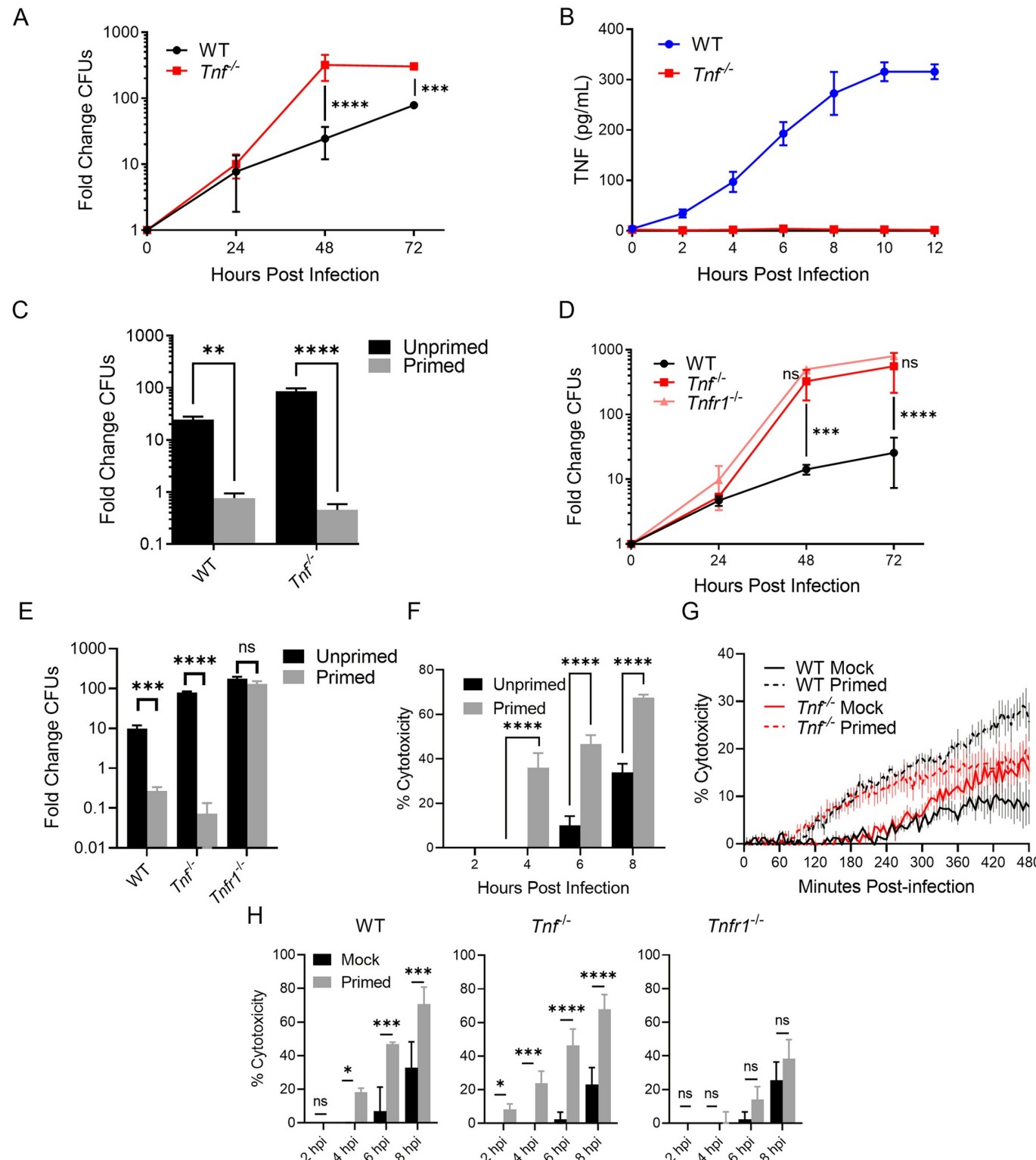

**Fig 1. TNFR1 signaling is required for restriction of L. pneumophila replication in macrophages and licenses cells to rapidly undergo cell death. (A,C-E)** WT, *Tnf*[-/-], (A, C) and *Tnfr1*[-/-] (D-E) BMDMs were infected with Δ*flaA L. pneumophila* at MOI = 1. (A,D) The fold change in CFUs were quantified at 24, 48, and 72 hours post-infection. (C,E) BMDMs were either primed with 10 ng/mL recombinant murine TNF or PBS mock control for 16 hours and infected for 48 hours. **(B)** WT and *Tnf*[-/-] BMDMs were infected with non-replicating Δ*flaA L. pneumophila* at MOI = 10 for 12 hours. Cytokine release was measured every 2 hours using ELISA. **(F-H)** WT, *Tnf*[-/-] (G, H), and *Tnfr1*[-/-] (H) BMDMs were infected with non-replicating Δ*flaA L. pneumophila* at MOI = 10 (E) or 50 (F-H) for 8

hours. Cytotoxicity was measured using LDH release assay (F,H) or PI uptake assay (G). BMDMs were primed with 10 ng/mL recombinant murine rTNF or PBS mock control for 16 hours. Graphs show the mean ± SEM of triplicate wells. * is p<0.05, ** is p<0.01, *** is p<0.001, and **** is p<0.0001 by 2-way ANOVA with Šídák's post-test (A-D, F-H) or Student's t-test (E). NS is not significant. Data shown are representative of at least three independent experiments.

responsible for mediating restriction of *L. pneumophila* at later timepoints. Indeed, infection of BMDMs with Δ*flaA L. pneumophila* resulted in gradual secretion of TNF, plateauing at about 10 to 12 hours post-infection (Fig 1B). To mimic this endogenous TNF produced following initial infection and subsequent restriction, we primed cells with recombinant TNF (rTNF) for 16 hours prior to infection. The bacterial replication observed in WT BMDMs was completely abrogated when cells were primed with rTNF (Fig 1C). Likewise, rTNF priming limited bacterial replication in *Tnf*[-/-] BMDMs.

TNF can signal through either TNF receptor 1 (TNFR1) or 2 (TNFR2), and each are known to contribute positively to host defense during pulmonary *L. pneumophila* infection, with TNFR1 promoting an inflammatory environment and restriction of *L. pneumophila* infection within the lung and TNFR2 limiting immunopathology [27,56]. We infected *Tnfr1*[-/-] and *Tnfr1*[-/-]*Tnfr2*[-/-] BMDMs, as well as WT and *Tnf*[-/-] BMDMs as controls. Bacterial replication was greatly increased in the absence of TNFR1, consistent with our observation in *Tnf*[-/-] BMDMs (Fig 1D). Crucially, priming with rTNF was able to restrict bacterial replication in *Tnf*[-/-] BMDMs, but did not restrict bacterial replication in *Tnfr1*[-/-] BMDMs (Fig 1E). Doubly deficient *Tnfr1*[-/-]*Tnfr2*[-/-] BMDMs demonstrated no additional defect in bacterial control relative to singly deficient *Tnfr1*[-/-] BMDMs (S1A Fig). Thus, TNF signals chiefly through TNFR1 to mediate restriction of *L. pneumophila* replication in BMDMs, in agreement with previous findings [27].

We then sought to determine whether TNF signaling restricts *L. pneumophila* infection via increased production of anti-microbial molecules. TNF signaling through NF-κB and MAPK pathways upregulates production of anti-microbial molecules such as reactive nitrogen and oxygen species (RNS, ROS) [24,57]. BMDMs lacking inducible nitric oxide synthase (*Nos2*[-/-]), the enzyme responsible for reactive nitrogen production, showed no defect in control of *L. pneumophila* replication relative to WT BMDMs (S1B Fig). We also observed that BMDMs deficient in NADPH oxidase 2 (*Cybb*[-/-]), the enzyme responsible for reactive oxygen production in the lysosome, did not show a defect in controlling *L. pneumophila* replication, in agreement with previous findings [27]. We additionally observed no defect in TNF secretion in *Nos2*[-/-] or *Cybb*[-/-] cells (S1C Fig). While we saw no basal defect in control of *L. pneumophila* replication in the absence of NADPH oxidase-derived ROS, we did observe a decreased ability of *Cybb*[-/-] cells to restrict *L. pneumophila* following exogenous TNF priming (S1D Fig). This is in line with studies that have shown a role for ROS and TNF in control of *L. pneumophila* infection [27]. Even in the absence of NADPH oxidase, however, we observed a significant decrease in bacterial growth in the context of TNF priming (S1D Fig). These data suggested that neither RNS nor NADPH oxidase-derived ROS are entirely responsible for TNFR1-mediated control of *L. pneumophila* replication and prompted us to investigate alternative mechanisms of TNF-mediated restriction.

Cell-extrinsic TNF signaling frequently mediates cell death in response to foreign insults [58]. Cell death in response to infection can restrict intracellular bacterial replication by limiting the replicative niche of the pathogen [38]. We therefore sought to determine whether TNF promotes cell death during *L. pneumophila* infection. To avoid any confounding effect of bacterial replication, we infected cells with thymidine auxotrophic bacteria incapable of replicating in the absence of exogenous thymidine. We measured cell death by monitoring both release of the cytosolic protein lactate dehydrogenase (LDH) into the extracellular space, as

well as uptake of the membrane-impermeable dye propidium iodide (PI). We observed LDH release (Fig 1F) and PI uptake (Fig 1G) in *L. pneumophila*-infected WT and *Tnf*[-/-] BMDMs by 6–8 hours following infection. WT and *Tnf*[-/-] BMDMs, when primed with rTNF prior to *L. pneumophila* infection, instead demonstrated hallmarks of death as early as 2–4 hours post infection, a significant acceleration relative to unprimed controls (Fig 1F and 1G). In accordance with our data indicating a requirement for TNFR1 signaling in control of bacterial restriction, we additionally observed that accelerated death in rTNF-primed cells was reliant on TNFR1 signaling (Fig 1H). These data indicate that TNF priming through TNFR1 licenses cells to rapidly undergo cell death and limit bacterial replication in response to *L. pneumophila* infection.

## The *Legionella* type IV secretion system triggers TNF-licensed activation of the caspase-1 and -11 inflammasomes

Infection of macrophages by *L. pneumophila* and subsequent injection via the T4SS results in assembly and activation of the inflammasome in the host cell cytosol [20,33,55,59]. Cytosolic recognition of *L. pneumophila* flagellin specifically is mediated by the NAIP5/NLRC4 inflammasome, and results in efficient restriction of intracellular replication [33,55]. Among the downstream effects of inflammasome activation are the processing of IL-1 family cytokines and the assembly of gasdermin pores to execute an inflammatory form of cell death known as pyroptosis [60]. Even during infection with flagellin-deficient *L. pneumophila*, which does not induce NAIP5/NLRC4 activation, we observed significant induction of *L. pneumophila*-triggered cell death following TNF priming (Fig 1F–1H). Importantly, using a *ΔflaAΔdotA* strain, we found that PI uptake and LDH release following *L. pneumophila* infection is entirely dependent on the T4SS, indicating that rapid cell death following TNF priming requires detection of T4SS activity (Fig 2A and S2 Fig).

We therefore sought to determine what form of cell death is potentiated by TNF priming upon *L. pneumophila* infection. Cell death downstream of inflammasome detection of bacterial ligands is potentiated by inflammatory caspases. In mice, caspase-1 is activated within a canonical inflammasome that typically contains an NLR and an adaptor protein [30–32]. Caspase-11, meanwhile, is activated within a noncanonical inflammasome in response to direct sensing of cytosolic LPS [37,61]. Notably, we observed that, within the first 8 hours of infection, TNF-licensed cell death as measured by PI uptake and LDH release depended on caspases-1 and -11 (Fig 2B and 2C). Likewise, we observed that release of IL-1 family cytokines following *L. pneumophila* infection was significantly increased in the context of TNF priming (Fig 2D). However, in the absence of both caspases-1 and -11, we found that IL-1 release was severely attenuated (Fig 2D). These data suggest that TNF licenses cells to undergo pyroptosis and secrete IL-1 family cytokines rapidly in response to *L. pneumophila* T4SS activity in a caspase-1/11 dependent manner.

We then examined whether caspase-1 and -11 are required for TNF-mediated restriction of bacterial replication in BMDMs. We observed that, similarly to WT and *Tnf*[-/-] BMDMs, *Casp1*[-/-]*Casp11*[-/-] BMDMs more robustly restrict bacterial replication when primed with rTNF (Fig 2E). It is worth noting that TNF-mediated restriction is slightly less robust in *Casp1*[-/-]-*Casp11*[-/-] BMDMs relative to WT or *Tnf*[-/-] BMDMs, though not to the point of statistical significance. In addition, we observed that at later timepoints, *Casp1*[-/-]*Casp11*[-/-] BMDMs still underwent cell death, as measured by PI uptake (Fig 2C). These data suggest that, while TNF licenses cells to rapidly engage caspase-1/11-mediated death, IL-1 release, and restriction of *L. pneumophila* infection, other caspase-1/11-independent factors contribute to TNF-mediated cell death and restriction of bacterial replication.

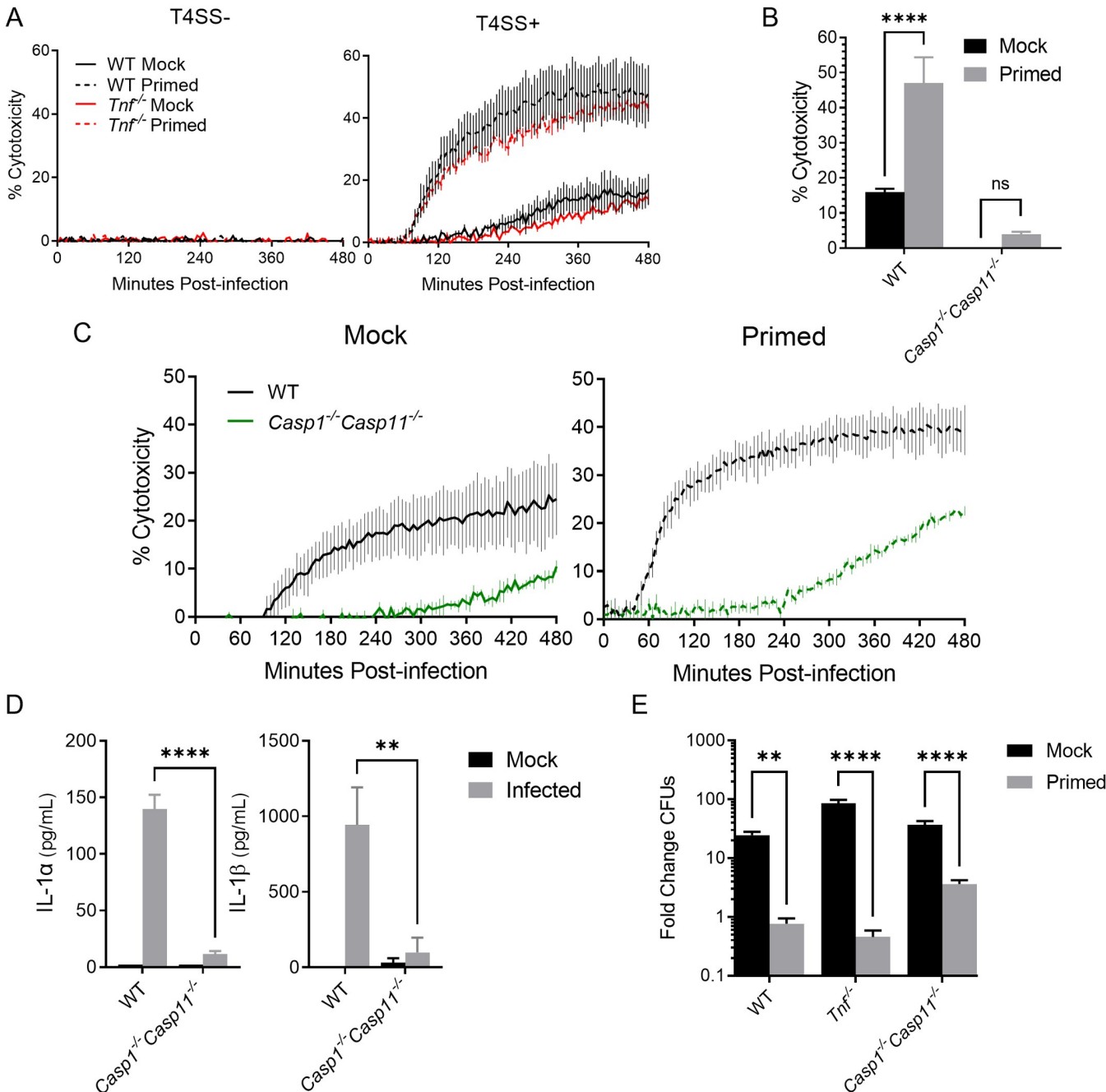

**Fig 2. *Legionella* type IV secretion system activity triggers TNF-licensed activation of the caspase-1 and -11 inflammasomes. (A)** WT and *Tnf*[-/-] BMDMs were primed with either 10 ng/mL rTNF or PBS mock control for 16 hours prior to infection with non-replicating Δ*flaA L. pneumophila* or Δ*dotA* bacteria lacking a functional T4SS at MOI = 10. **(B-C)** WT and *Casp1*[-/-]*Casp11*[-/-] BMDMs were primed with either 10 ng/mL rTNF or PBS mock control for 16 hours and then infected for 8 hours with non-replicating Δ*flaA L. pneumophila* at MOI = 50 (B) or 10 (C). Cytotoxicity was measured by PI uptake assay (A,C) or LDH release assay (B). **(D)** WT and *Casp1*[-/-]*Casp11*[-/-] BMDMs were primed with 10 ng/mL rTNF for 16 hours then left uninfected or infected with non-replicating Δ*flaA L. pneumophila* at MOI = 50. Cytokine release measured by ELISA. **(E)** WT, *Tnf*[-/-], and *Casp1*[-/-]*Casp11*[-/-] BMDMs were primed with either 10 ng/mL rTNF or PBS mock control for 16 hours prior to infection with replicating Δ*flaA L. pneumophila* at MOI = 1. Fold change in CFUs was quantified at 72 hours post-infection. Graphs show the mean ± SEM of triplicate wells. * is $p<0.05$, ** is $p<0.01$, *** is $p<0.001$, and **** is $p<0.0001$ by 2-way ANOVA with Šídák's post-test (A-D) or Tukey HSD (E). NS is not significant. Data shown are representative of at least three independent experiments.

## TNF priming licenses cells to activate the non-canonical inflammasome during *L. pneumophila* infection

Given the pyroptotic nature of TNF-licensed cell death we observed in response to non-flagellated *L. pneumophila*, we sought to determine which inflammasomes are responsible. The NLRP3 inflammasome oligomerizes in response to diverse cytosolic triggers, including reactive oxygen species, potassium efflux, and lysosomal damage signals [38,62], and contributes to the inflammasome response to *L. pneumophila* in the absence of TNF priming [36,37]. We observed significant attenuation of IL-1β release in MCC950-treated and TNF-primed BMDMs infected with *L. pneumophila*, indicating that the NLRP3 inflammasome is required for IL-1β release in TNF-primed cells (Fig 3A). Notably, we observed no decrease in IL-1α release or in cell death in TNF-primed BMDMs treated with the NLRP3-specific inhibitor, MCC950, compared to mock-treated cells (Fig 3A and 3B). These data suggest that the NLRP3 inflammasome contributes to IL-1β release but is not required for cell death and IL-1α release in TNF-licensed cells, indicating that another inflammasome is activated in TNF-primed BMDMs following *L. pneumophila* infection.

The non-canonical inflammasome, formed by caspase-11, is able to directly mediate pyroptosis and release of IL-1α, but cannot independently process IL-1β, and thus requires secondary NLRP3 activation to mediate release of active IL-1β during *L. pneumophila* infection following LPS-priming [36,37]. We thus tested whether the non-canonical inflammasome is involved in the TNF-licensed response to *L. pneumophila*. Indeed, we observed a significant decrease in TNF-licensed cell death early during infection, as well as significant attenuation of both IL-1α and IL-1β release, in *Casp11*$^{-/-}$ BMDMs relative to wild-type controls (Fig 3C–3E). These data implicate the caspase-11 non-canonical inflammasome as a downstream target of TNF licensing in *L. pneumophila* infection.

We next sought to determine how TNF may be enhancing caspase-11 inflammasome activation during infection. Activation of NF-κB and MAP kinase signaling and subsequent gene expression, as well as caspase-8-mediated inflammatory gene expression, are mediated by upstream TNF receptor signaling [3,6,41,63,64]. We hypothesized that TNF licensing of non-canonical inflammasome activation involves upregulated expression of non-canonical inflammasome components such as caspase-11, gasdermin D, and IL-1 family cytokines. We used quantitative RT-PCR to determine the effect of TNF priming on transcription of inflammasome factors prior to and during *L. pneumophila* infection. As early as 2 hours following TNF treatment, we observed statistically significant increases in *Il1a*, *Il1b*, and *Il18* mRNA levels (Fig 3F). We additionally observed increased expression of *Casp11*, though this increase did not achieve statistical significance (Fig 3F). TNF-mediated increase in *Casp11* transcription can still be observed at 16 hours following priming (at the time of infection), while *Il1a* and *Il1b* transcription fades by that time (S3A Fig). Additional inflammasome-related genes, such as *Casp8* and *Casp1*, did not show an increase in mRNA expression in TNF-primed cells (S3B Fig). We observed a substantial increase in the level of caspase-11 protein in TNF-primed cellular lysates prior to *L. pneumophila* infection, suggesting that TNF licenses cells to react more rapidly to infection by upregulating caspase-11 levels (Fig 3G). Accordingly, TNF priming also resulted in release of cleaved caspase-11 into the supernatant of infected cells (Fig 3G). These data collectively indicate that TNF priming leads to increased caspase-11 expression and activation during infection.

As the non-canonical inflammasome detects cytosolic LPS, we aimed to elucidate how *L. pneumophila* is being detected by the non-canonical inflammasome, and whether this detection itself is enhanced by TNF priming. Guanylate Binding Proteins (GBPs) are activated downstream of IFN signaling, subsequently localizing to and permeabilizing pathogen-

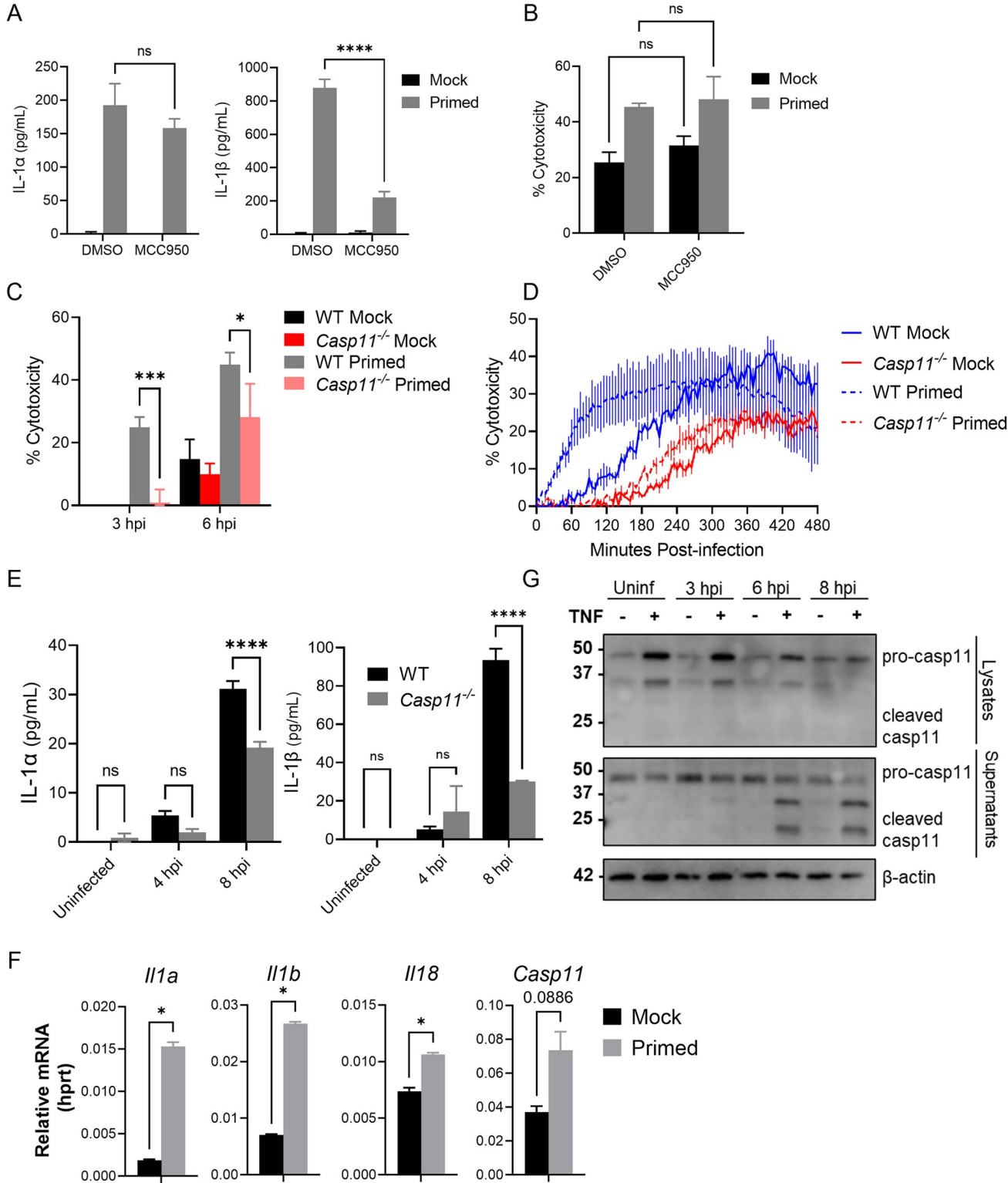

**Fig 3. TNF priming licenses cells to activate the non-canonical inflammasome during *L. pneumophila* infection. (A-B)** *Tnf⁻/⁻* BMDMs were either primed with 10 ng/mL rTNF or PBS mock control for 16 hours prior to infection with non-replicating Δ*flaA L. pneumophila* at MOI = 50 for 6 hours. Cells were additionally either treated with DMSO vehicle or 1 uM MCC950 for 1 hour prior to infection. Cytokine release was measured by ELISA and cytotoxicity was determined by LDH release assay. **(C-D)** WT and *Casp11⁻/⁻* BMDMs primed with either 10 ng/mL rTNF or PBS mock control for 16 hours prior to infection with non-replicating Δ*flaA L. pneumophila* at MOI = 50 (C) or 10 (D). **(E)** WT and *Casp11⁻/⁻* BMDMs were primed with 10 ng/mL

rTNF for 16 hours prior to infection with non-replicating Δ*flaA L. pneumophila* at MOI = 50. Cytokine release was measured by ELISA. **(F)** WT BMDMs were treated with either 10 ng/mL rTNF or PBS for 2 hours, then supernatants were collected and cells lysed for immunoblot and qPCR. **(G)** WT BMDMs were primed with either 10 ng/mL rTNF or PBS mock control for 16 hours prior to infection with non-replicating Δ*flaA L. pneumophila* at MOI = 50. Supernatants were collected and BMDDMs lysed at times indicated. Graphs show the mean ± SEM of triplicate wells. * is p<0.05, ** is p<0.01, *** is p<0.001, and **** is p<0.0001 by 2-way ANOVA with Šídák's post-test (A-E) or Student's t-test (F). NS is not significant. Data shown are representative of at least two independent experiments.

containing vacuoles in order to introduce bacterial LPS to the cytosol for caspase-11 detection, as well as binding bacterial surfaces directly [65,66]. GBPs promote inflammasome responses to *L. pneumophila* in mouse macrophages downstream of IFN signaling [67,68]. However, BMDMs from *Gbp*^Chr3-/- mice deficient for the six GBPs found on murine chromosome 3 did not exhibit a significant loss of TNF-licensed cell death (S4A Fig) or bacterial restriction (S4B Fig). This suggests that in the context of TNF priming, *L. pneumophila* activates caspase-11 independently of the GBPs on chromosome 3. Altogether, these data indicate that TNF priming licenses BMDMs to engage the caspase-11-mediated non-canonical inflammasome by upregulating inflammasome components in advance of infection.

## Caspase-8 contributes to cell death independently of its autocleavage downstream of TNF signaling

While TNF-licensed cell death is significantly attenuated in the absence of caspase-1 and caspase-11, we observed that, even in the absence of these caspases, TNF-primed macrophages exhibited cell death at later timepoints (Fig 2C). Prior studies revealed that other inflammasome stimuli can induce cells to undergo delayed pyroptosis that requires caspase-8 in the absence of canonical pyroptotic mediators [50,51,69,70]. We therefore investigated the possible role of caspase-8 in compensatory or redundant mechanisms of cell death downstream of TNF signaling in *L. pneumophila* infection. TNF triggers caspase-8 mediated cell death in the absence or inhibition of NF-κB-mediated survival signals [3–6]. Thus, we sought to determine whether TNF priming leads to caspase-8-dependent cell death following *L. pneumophila* infection. Mice deficient in caspase-8 signaling experience uncontrolled activation of RIPK3-mediated necroptosis, a form of cell death that eliminates cells lacking caspase-8 activity, and thus are embryonically lethal [44–46,71]. To address the role of caspase-8, we therefore used BMDMs lacking both RIPK3 and CASP8. BMDMs from *Ripk3*^-/-*Casp8*^-/- mice exhibited significantly less TNF-licensed cell death following *L. pneumophila*-infection relative to *Ripk3*^-/- or WT controls (Fig 4A). We then aimed to determine whether the caspase-8-mediated death we observe is necessary for restriction of *L. pneumophila* replication. Much like in caspase-1/11 deficiency, we still observed robust TNF priming-mediated restriction of bacterial replication in the absence of caspase-8 (Fig 4B). Similarly to *Casp1*^-/-*Casp11*^-/- BMDMs, we observed less robust TNF-mediated restriction in *Ripk3*^-/-*Casp8*^-/- BMDMs relative to controls, though not to the point of statistical significance. These data suggest that TNF signaling poises cells to engage caspase-8-mediated cell death in response to *L. pneumophila* infection, and that this pathway contributes to but is not required for bacterial restriction.

We then sought to dissect which downstream pathways are engaged by caspase-8 in TNF-primed BMDMs to mediate cell death following infection. During extrinsic apoptosis, caspase-8 is recruited by RIPK1, at which point caspase-8 dimerizes and auto-cleaves before cleaving downstream apoptotic substrates caspase-3 and caspase-7 [72]. Independently of auto-cleavage, caspase-8 also regulates inflammatory gene expression downstream of TLR signaling [43,47–49]. Caspase-8 can additionally compensate for caspase-1 in the NAIP/NLRC4 and NLRP3 inflammasomes, though it is unknown whether this is dependent on auto-cleavage

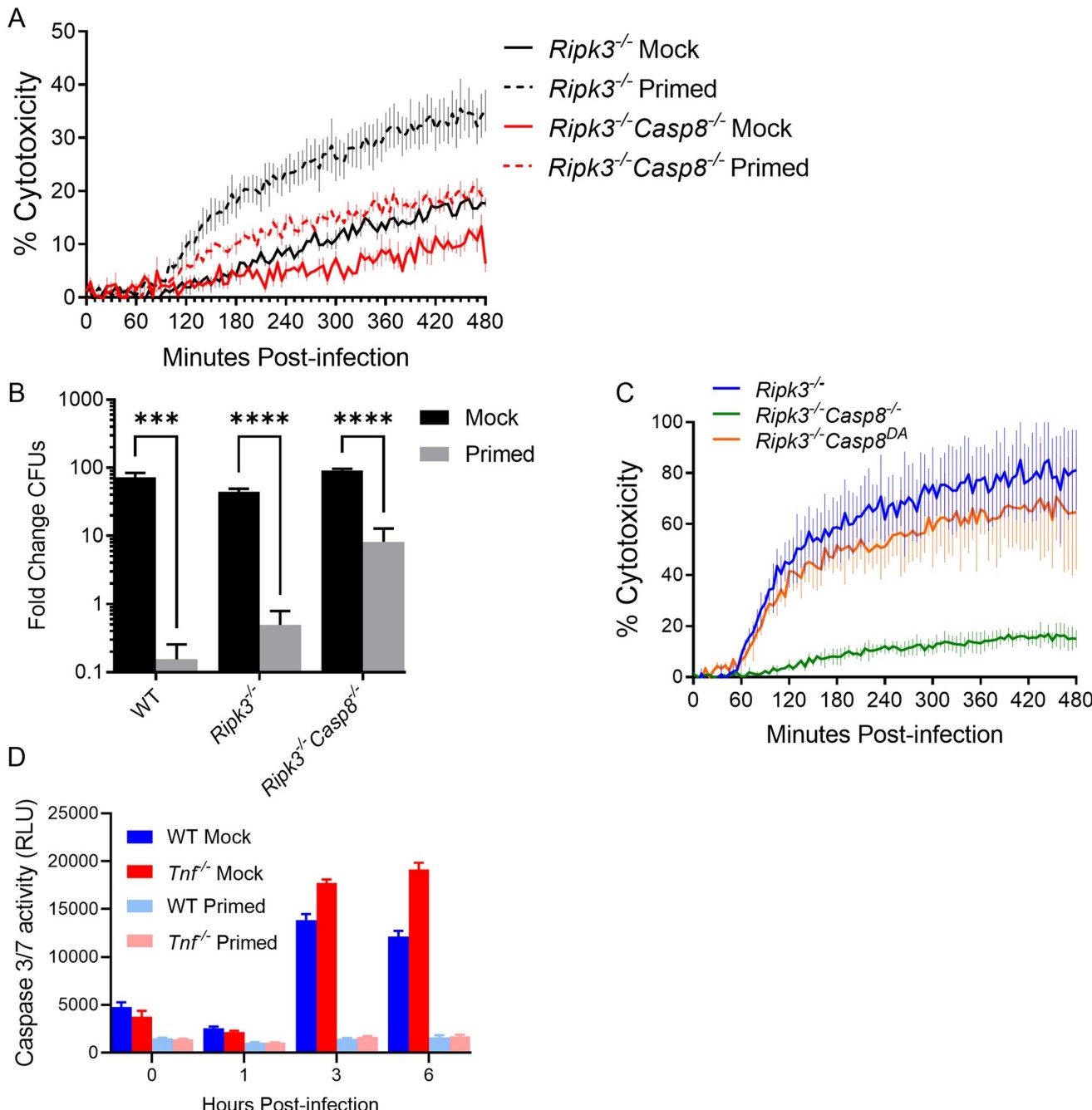

**Fig 4. Caspase-8 mediates optimal cell death independently of its autocleavage downstream of TNF signaling. (A-B)** WT (B), *Ripk3^-/-^*, *Ripk3^-/-^Casp8^-/-^*, and *Ripk3^-/-^Casp8^DA^* BMDMs were primed with either 10 ng/mL rTNF or PBS mock control for 16 hours prior to infection with non-replicating (A) or replicating (B) Δ*flaA L. pneumophila* at MOI = 10 (A) or 1 (B) for 8 (A) or 72 (B) hours. **(C)** *Ripk3^-/-^*, *Ripk3^-/-^Casp8^-/-^*, and *Ripk3^-/-^Casp8^DA^* BMDMs were primed with 10 ng/mL rTNF for 16 hours prior to infection with non-replicating Δ*flaA L. pneumophila* at MOI = 10 for 8 hours. **(D)** BMDMs were primed with either 10 ng/mL rTNF or PBS mock control for 16 hours prior to infection with non-replicating Δ*flaA L. pneumophila* at MOI = 50 for 6 hours. Caspase-3/7 activity was measured with the caspase GLO assay. Graphs show the mean ± SEM of triplicate wells. * is $p<0.05$, ** is $p<0.01$, *** is $p<0.001$, and **** is $p<0.0001$ by Student's t-test. NS is not significant. Data shown are representative of at least three independent experiments.

[49,70,73]. We observed a significant defect in IL-1 family cytokine release in the absence of caspase-8 in TNF-primed BMDMs (S4B and S5A Figs), suggesting a role for caspase-8 in TNF-licensed inflammatory cytokine release. To investigate the contribution of caspase-8-mediated inflammatory gene expression downstream of TNF, we assessed expression of caspase-11 in the absence of caspase-8. Caspase-11 remained more highly expressed in TNF-primed *Ripk3^-/-Casp8^-/-* BMDMs relative to unprimed BMDMs, suggesting that caspase-8 is not required for TNF-mediated upregulation of caspase-11 (S5B Fig).

To test the specific contribution of caspase-8 autoprocessing in TNF-mediated cell death and restriction of *L. pneumophila* replication, we used knock-in mutant mice bearing a D387A mutation in caspase-8 (*Ripk3^-/-Casp8^DA*), which eliminates caspase-8 autoprocessing and apoptotic activity [43,71,74]. Infection of TNF-primed *Ripk3^-/-Casp8^DA* BMDMs resulted in substantial increase in cell death, as measured by PI uptake, relative to unprimed *Ripk3^-/-Casp8^DA* BMDMs. Furthermore, the levels of cell death were comparable to TNF-primed *Ripk3^-/-* cells, whereas we observed substantially lower cell death in both TNF-primed and unprimed *Ripk3^-/-Casp8^-/-* cells (Fig 4C and S6 Fig). These data suggest that caspase-8 is required for optimal cell death during *L. pneumophila* infection, but that caspase-8 autoprocessing, and thus the classical apoptotic function of caspase-8, is not required. Indeed, using a fluorogenic substrate assay to interrogate downstream caspase-3/7 enzymatic activity during infection, we found that TNF-licensed cells demonstrated significantly lower caspase-3 and -7 (Fig 4D) activity relative to unprimed cells. These data indicate that an additional, non-apoptotic mechanism of TNF-licensed death is therefore downstream of caspase-8 in *L. pneumophila* infection.

## TNF priming licenses gasdermin-dependent cell death downstream of caspase-1, caspase-8, and caspase-11 following infection

Observing a requirement for multiple caspases in TNF-licensed death following *L. pneumophila* infection, we next investigated the role of terminal gasdermins in this death. Inflammasome activation triggers caspase-mediated cleavage of the cytosolic pore-forming protein gasdermin D. This frees the N-terminal pore-forming domain from the autoinhibitory C-terminal domain, allowing oligomerization and formation of a pyroptotic pore from which the cell can release inflammatory cytokines and intracellular damage-associated molecular patterns [53,60]. To dissect the role of gasdermins in TNF-licensed death, we used BMDMs deficient in either gasdermin D (*Gsdmd^-/-*) or the closely related protein, gasdermin E (*Gsdme^-/-*), which is known to be activated by caspase-8 downstream of TNF to cause pyroptosis in certain contexts and cell types [75]. We additionally infected BMDMs lacking both gasdermin D and gasdermin E (*Gsdmd^-/-Gsdme^-/-*), as gasdermin E has been suggested to mediate secondary or non-canonical pyroptosis in the absence of gasdermin D [51,75]. We observed no decrease in TNF-licensed cell death in *Gsdme^-/-* BMDMs, confirming that gasdermin E does not bear a primary role in cell death following *L. pneumophila* infection (Fig 5A). We did, however, observe a significant defect in cell death in both *Gsdmd^-/-* and *Gsdmd^-/-Gsdme^-/-* macrophages, suggesting that gasdermins are required for TNF-licensed cell death during *L. pneumophila* infection (Fig 5A and 5B). While we did observe a significant attenuation of cell death solely in the absence of gasdermin D, we observed a further significant decrease in the absence of both gasdermins D and E, suggesting some compensation by gasdermin E in the absence of gasdermin D (Fig 5A). This is in keeping with a recent study that found a compensatory role for gasdermin E in pyroptosis downstream of caspase-8, specifically in the context of gasdermin D deficiency [75]. Later, at 8 hours post-infection, we observed a delayed return of cell death, which may be reflective of compensatory apoptosis in the absence of pyroptotic death (Fig 5B). We also observed significant abrogation of IL-1α and IL-1β release in the absence of gasdermins D

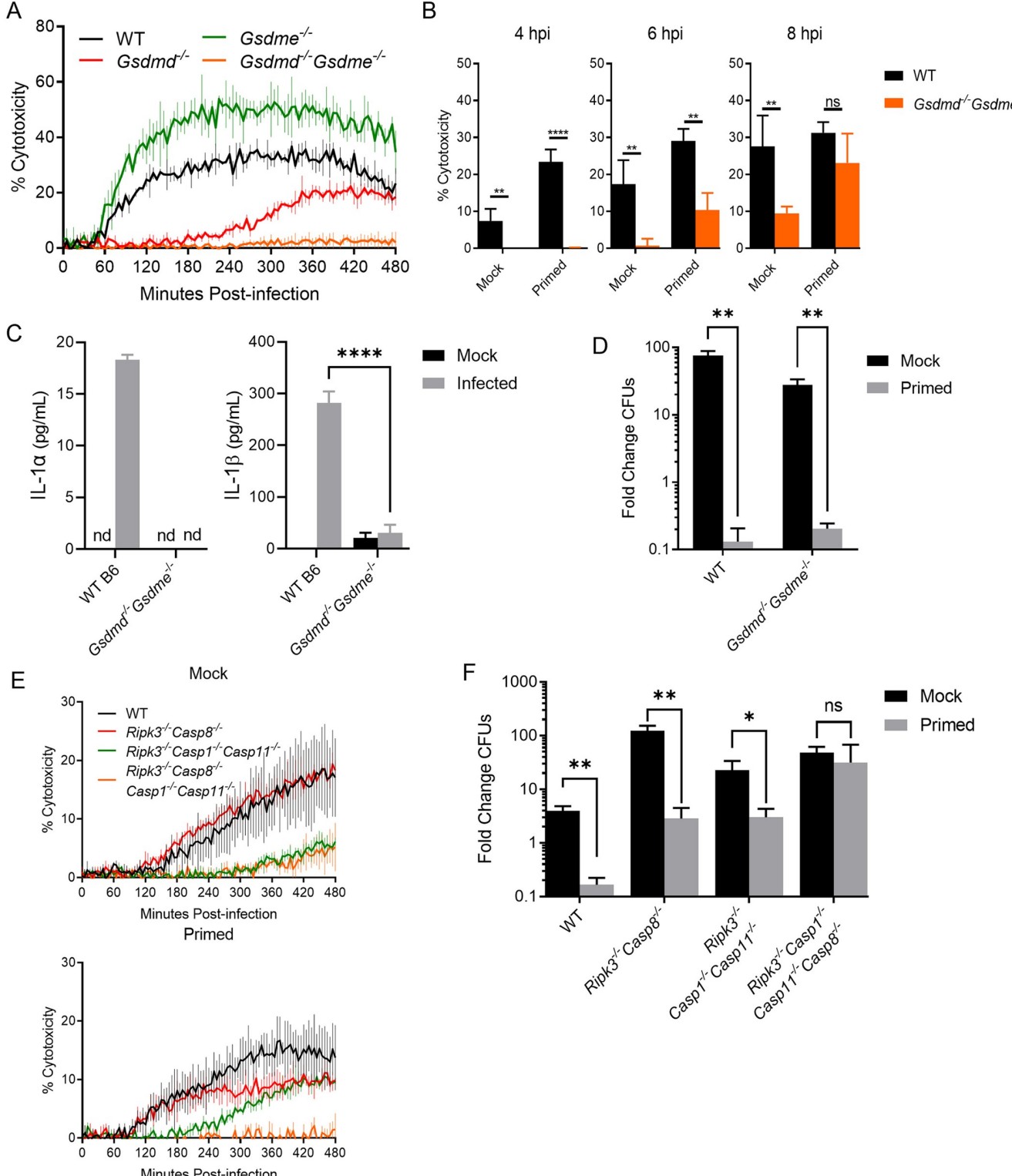

**Fig 5. TNF priming licenses a gasdermin-dependent cell death, requiring caspase-1, caspase-8, and caspase-11 following infection. (A)** WT, *Gsdme*[−/−], *Gsdmd*[−/−], and *Gsdmd*[−/−]*Gsdme*[−/−] BMDMs were primed with either 10 ng/mL rTNF or PBS mock control for 16 hours prior to infection with non-replicating Δ*flaA L. pneumophila* at MOI = 10. Cytotoxicity was measured by PI uptake assay. **(B)** WT and *Gsdmd*[−/−]*Gsdme*[−/−] BMDMs were primed with either 10 ng/mL rTNF or PBS mock control for 16 hours prior to infection with non-replicating Δ*flaA L. pneumophila* at MOI = 50, and cells. Cytotoxicity was measured by LDH release assay. **(C)** WT and *Gsdmd*[−/−]*Gsdme*[−/−] BMDMS were primed with 10 ng/mL rTNF for 16 hours prior to infection with non-replicating Δ*flaA L.*

*pneumophila* at MOI = 50 for 6 hours. Cytokine release was measured by ELISA. **(D)** WT and *Gsdmd$^{-/-}$Gsdme$^{-/-}$* BMDMs were primed with either 10 ng/mL rTNF or PBS mock control for 16 hours prior to infection with replicating Δ*flaA L. pneumophila* at MOI = 1 for 72 hours. **(E-F)** *Ripk3$^{-/-}$*, *Ripk3$^{-/-}$Casp8$^{-/-}$*, *Ripk3$^{-/-}$Casp1$^{-/-}$Casp11$^{-/-}$*, and *Ripk3$^{-/-}$Casp1$^{-/-}$Casp11$^{-/-}$Casp8$^{-/-}$* BMDMs were primed with either 10 ng/mL rTNF or PBS mock control for 16 hours prior to infection with non-replicating (E) or replicating (F) Δ*flaA L. pneumophila* at MOI = 10 (E) or 1 (F). Graphs show the mean ± SEM of triplicate wells. * is p<0.05, ** is p<0.01, *** is p<0.001, and **** is p<0.0001 by 2-way ANOVA with Šídák's post-test (B,C) or Student's t-test (D,E). NS is not significant. Data shown are representative of at least three independent experiments.

and E following infection during TNF priming, indicating that TNF licenses cells to undergo gasdermin-mediated pyroptosis to release IL-1 cytokines (Fig 5C). Of note, cells deficient for both gasdermin D and E still exhibited complete abrogation of bacterial growth in the context of TNF priming, suggesting additional TNF-mediated protective mechanisms independent of gasdermin D and E (Fig 5D).

These data provoked the hypothesis that TNF-licensed caspase-1, caspase-8, and caspase-11 may all be participating in inflammasome activation to mediate subsequent pyroptosis in *L. pneumophila*-infected macrophages. To further define the relative contributions of caspases-1, -8, and -11, we generated BMDMs from *Ripk3$^{-/-}$*, *Ripk3$^{-/-}$Casp8$^{-/-}$*, *Ripk3$^{-/-}$Casp1$^{-/-}$Casp11$^{-/-}$*, and *Ripk3$^{-/-}$Casp8$^{-/-}$Casp1$^{-/-}$Casp11$^{-/-}$* mice. Using these cells, we observed that TNF-licensed cell death was partially attenuated in the absence of either caspase-8 or caspases-1 and -11 (Fig 5E). However, TNF-licensed cell death was fully abrogated in the absence of all three caspases (Fig 5E). We likewise assessed restriction of bacterial replication in these cell populations and found that, while the absences of caspase-8 or caspases-1 and -11 still allowed for a moderate but significant TNF-mediated attenuation of bacterial growth, the absence of all three caspases rendered cells unable to restrict *L. pneumophila* replication even in the context of TNF priming (Fig 5F). These data together suggest that in TNF-licensed BMDMs infected with *L. pneumophila*, caspases-1, -8, and -11 collectively facilitate gasdermin-dependent, pyroptotic cell death and control of intracellular bacterial replication.

## TNFR1 and caspase-8 are required for control of pulmonary *L. pneumophila* infection

The TNF-mediated multi-caspase control of *L. pneumophila* we observed in BMDMs may contribute to the effective TNF-dependent control of bacterial replication *in vivo*. We therefore investigated whether the TNF and caspase-8-mediated restriction of flagellin-deficient *L. pneumophila* in primary BMDMs *in vitro* could also be observed during active pulmonary infection. In keeping with our findings in cell culture, we observed that *Tnf$^{-/-}$* mice were unable to control bacterial loads in the lung, unlike their wild type C57BL/6 counterparts (Fig 6A). Likewise, we observed a significant defect in bacterial control in *Tnfr1$^{-/-}$* mice compared to WT, suggesting a role for TNF signaling through TNFR1 in control of pulmonary *L. pneumophila* infection (Fig 6B). These data agree with our findings *in vitro*, as well as other *in vivo* studies [8,13,20,56].

Moreover, *Ripk3$^{-/-}$Casp8$^{-/-}$* mice also exhibited a significant defect in restriction of bacterial replication relative to WT and *Ripk3$^{-/-}$* control mice, indicating a requirement for caspase-8 in control of *L. pneumophila* infection *in vivo* (Fig 6C). Notably, we observed that *Ripk3$^{-/-}$Casp8$^{DA}$* mice unable to undergo caspase-8 autocleavage exhibited similar levels of bacterial replication compared to *Ripk3$^{-/-}$* mice or *Ripk3$^{-/-}$Casp8$^{DA/+}$* littermate controls (Fig 6D), in agreement with our *in vitro* findings indicating that caspase-8 autocleavage is not necessary for TNF-mediated restriction of *L. pneumophila* replication (Fig 4C). These data together indicate that signaling through TNFR1 and non-apoptotic caspase-8 activity mediate control of pulmonary *L. pneumophila* infection.

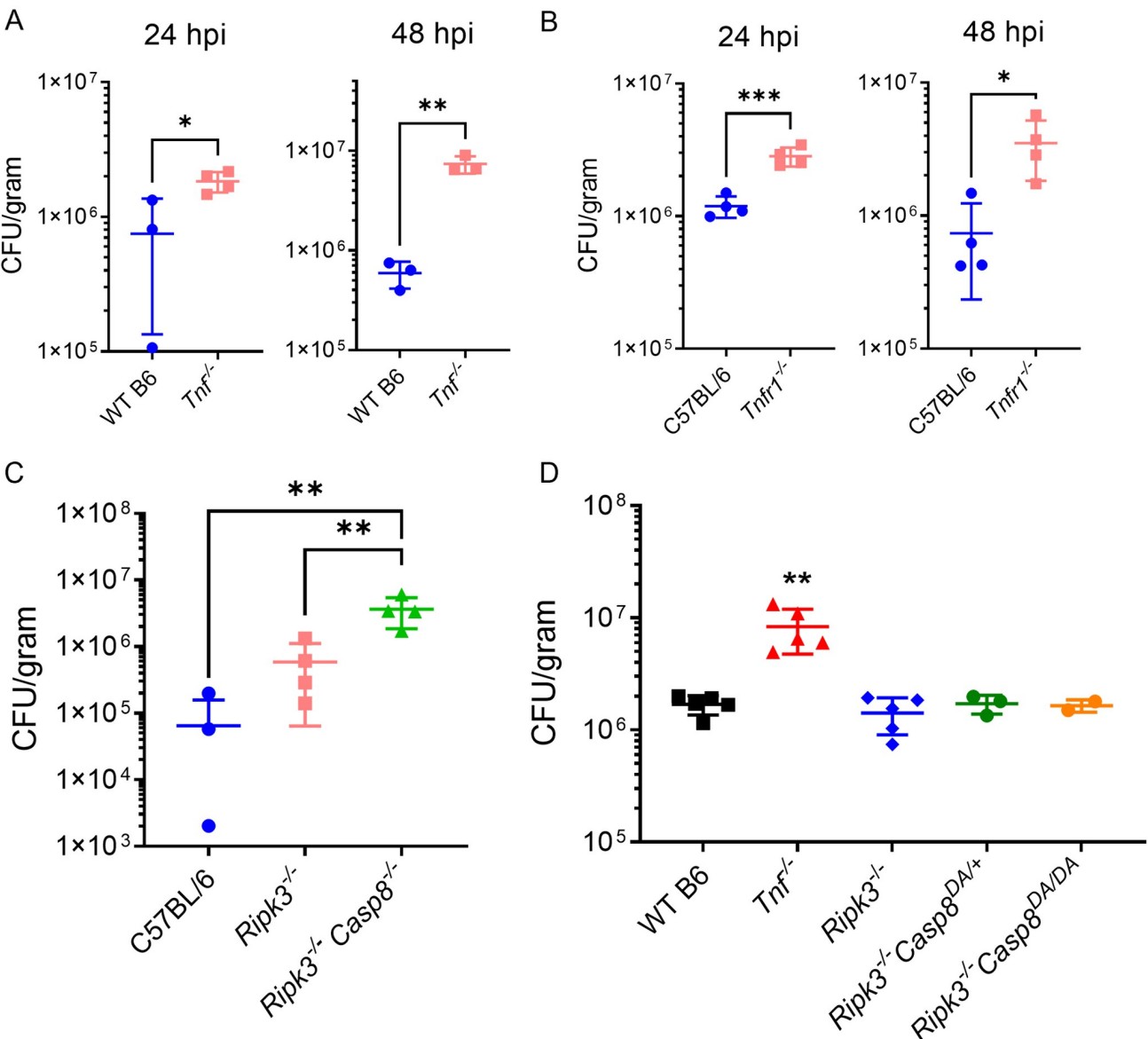

**Fig 6. TNFR1 and caspase-8 are required for control of pulmonary *L. pneumophila* infection.** WT, *Tnf*⁻/⁻ (A, D), *Tnfr1*⁻/⁻ (B), *Ripk3*⁻/⁻ (C, D), *Ripk3*⁻/⁻*Casp8*⁻/⁻ (C), *Ripk3*⁻/⁻*Casp8*^DA/+ (D), and *Ripk3*⁻/⁻*Casp8*^DA/DA mice were infected intranasally with 1x10⁶ Δ*flaA L. pneumophila* in 40 μL sterile PBS. Lung CFUs were quantified after 48 hours of infection. Graph shows the ± SEM of three or four infected mice per group. * is p<0.05, ** is p<0.01, and *** is p<0.001 by Student's t-test (A,B) or one-way ANOVA with Tukey HSD post-test (C,D). Data shown are representative of at least two independent experiments.

## Discussion

The inflammatory response generated by the innate immune system determines the ability of the mammalian host to efficiently clear bacterial infection. While the relative contributions of immune mechanisms such as inflammasomes and cytokines have been well characterized, the cell-intrinsic effects of cytokines such as TNF in restricting intracellular bacterial pathogens have remained mechanistically obscure. In this study, we set out to determine how TNF controls *L. pneumophila* infection. Specifically, we used priming of macrophages with recombinant TNF prior to infection in order to mimic the kinetics of initially infected cells inducing

TNF production in uninfected neighboring cells to defend against the second wave of infection. We also used *L. pneumophila* lacking flagellin to avoid triggering the NAIP/NLRC4 inflammasome, as our findings and previous studies indicated that TNF can mediate restriction of flagellin-deficient *L. pneumophila* (Fig 1) [27]. Using this priming and infection model alongside genetic tools, we demonstrate that TNF signaling through TNFR1 licenses *L. pneumophila*-infected cells to rapidly and robustly undergo cell death independently of flagellin and the NAIP/NLRC4 inflammasome. We find that TNF promotes accelerated inflammasome activation, gasdermin-dependent pyroptosis, and release of IL-1 family cytokines. TNF is able to license this pyroptotic death in part by upregulating components of the caspase-11 non-canonical inflammasome ahead of infection. Furthermore, we find that following TNF priming, NLRP3 inflammasome activation and caspase-8 non-apoptotic activity in parallel mediate maximal cytokine release and death of infected cells. We find that caspases-1, -11, and -8 are all required for maximal TNF-dependent cell death and control of *L. pneumophila* replication within macrophages. We also observe a defect in control of pulmonary *L. pneumophila* infection in the absence of caspase-8. Thus, our findings indicate that TNF licenses the host to respond to flagellin-deficient *L. pneumophila* infection by rapidly activating cell death pathways downstream of caspases-1, -11, and -8, thereby restricting bacterial replication. Our findings also indicate a critical role for caspase-8 in control of pulmonary *L. pneumophila* infection independent of the NAIP5/NLRC4 inflammasome.

Intracellular pathogens elicit inflammasome responses as the sanctity of the intracellular compartment is violated by infectious insult [76]. The exact nature of the inflammasome activated is determined by the type of ligand detected, be it an injected bacterial protein, effector or toxin activity, foreign nucleic acid, or bacterial cell wall components [77]. Additionally, the inflammatory context in which these foreign signals are detected will orient the sensitivity, magnitude, and character of the inflammatory response. *L. pneumophila* robustly activates the NAIP5/NLRC4 inflammasome within infected macrophages by injecting flagellin into the host cytosol [20,33,55,59], and this process requires TNF signaling for optimal restriction of *L. pneumophila* replication [20]. Our study uses flagellin-deficient *L. pneumophila* to highlight that TNF also licenses rapid activation of multiple inflammasomes to restrict bacterial infection independently of flagellin and NAIP5/NLRC4 inflammasome activation. Our data demonstrate that caspase-11, which detects cytosolic LPS, is crucial for the early TNF-licensed response to *L. pneumophila* infection. This caspase-11 activation results in the formation of gasdermin D (GSDMD) pores that lead to cell lysis and IL-1α release. Importantly, TNF priming also enhances NLRP3 inflammasome-dependent release of IL-1β. It is possible that this NLRP3 inflammasome activation may be downstream of caspase-11 activation, likely by virtue of caspase-11-dependent gasdermin pores facilitating $K^+$ efflux [36,37]. Furthermore, it is possible that both caspase-1 and caspase-8 are being recruited to the NLRP3 inflammasome [50,73,78]. Regardless, we demonstrate that this TNF-mediated inflammasome activation is dependent on the action of the *L. pneumophila* T4SS. While our data suggest that the activating signal is independent of the GBPs on chromosome 3, it is possible that one of the other 7 murine GBPs is mediating vacuolar permeabilization in response to the T4SS [68]. Alternatively, it is possible that either an uncharacterized *L. pneumophila* effector molecule or even the process of secretion system injection alerts the promiscuous NLRP3 inflammasome, triggering further feedforward loops among the other inflammasomes.

While we observe a requirement for GSDMD in optimal TNF-licensed cell death in *L. pneumophila*-infected macrophages, we see further attenuation of cell death in the absence of both GSDMD and the related protein gasdermin E (GSDME), indicating that both GSDMD and GSDME contribute to TNF-licensed cell death in response to *L. pneumophila* infection. Notably, TNF-primed *Gsdme*$^{-/-}$ BMDMs exhibited no defect in cell death, indicating that

GSDMD is primarily responsible for cell lysis in this context. GSDME is cleaved by caspase-3 downstream of caspase-8 in cancer cells, intestinal epithelial cells, and macrophages [75,79,80]. This cleavage results in both pyroptosis and permeabilization of mitochondria, enhancing intrinsic apoptosis [81]. In THP-1 macrophages, GSDME mediates IL-1β release and limited pyroptosis in response to nigericin treatment and *Salmonella* infection, as well as pyroptosis in the absence of GSDMD [75]. GSDME does not universally contribute to pyroptosis, however, as in the setting of pathogen-induced NF-κB blockade GSDME is activated by a caspase-8/3 pathway yet exhibits no role in macrophage lysis [82]. Additionally, a secondary form of pyroptosis can occur downstream of caspase-8 in the absence of GSDMD [51]. It therefore appears likely that the residual, GSDMD-independent death we observe in our system is mediated by GSDME compensation downstream of caspase-8 and caspase-3. Intriguingly, we observed that although TNF-primed cell death was attenuated in the absence of GSDMD and GSDME, there still remained delayed cell death and restriction of *L. pneumophila* replication, indicating that additional host factors downstream of caspases-1, -11, and -8 contribute to cell death and restriction of *L. pneumophila*. These additional host factors may include accelerated phagolysosomal fusion with the *Legionella*-containing vacuole, which has been observed as a downstream consequence of TNF signaling by other groups [27]; or TNF-induced macrophage necrosis, which has been shown to be mediated by mitochondrial ROS in the context of *Mycobacterium* infection [83]. Alternatively, gasdermin-independent restriction of *L. pneumophila* may be a result of compensatory caspase-8-mediated apoptosis [84], which is in line with the gasdermin-independent cell death we observe occurring late during infection. This is additionally congruous with multiple studies which have shown that, especially *in vivo*, apoptotic death can compensate for the lack of caspase-1-mediated death late during infection [50,84,85].

The potential compensation by GSDME indicates an additional node of caspase-8-mediated compensation in the TNF-primed inflammasome response against *L. pneumophila*. Our study demonstrates that in TNF-primed cells, caspases-1, -11, and -8 collectively mediate cell death and control of *L. pneumophila* infection. While caspase-8 is thought to predominantly initiate extrinsic apoptosis, caspase-8 has been shown to compensate in the absence of caspase-1 to cleave shared substrates, including IL-1β and GSDMD [50–53,86]. Notably, during *L. pneumophila* infection in the absence of TNF priming, NAIP5/NLRC4 inflammasome activation leads to caspase-1 and -8-mediated activation of GSDMD and caspase-7, respectively [70]. In contrast, caspases-1, -8, and -11 are dispensable for restriction of flagellin-deficient *L. pneumophila* in the absence of TNF priming [50]. However, in TNF-primed macrophages infected with flagellin-deficient *L. pneumophila*, we see involvement of all three caspases, as we find that in the absence of caspases-1 and -11, caspase-8 contributes to restriction of *L. pneumophila* infection within BMDMs. Our data also indicate that caspase-8 autocleavage is not involved in control of *in vitro* or *in vivo* infection, suggesting that caspase-8 is promoting cell death and bacterial restriction independently of its apoptotic activity. It is possible that in addition to caspase-8's role in cell death, it may contribute to transcription of inflammatory cytokine genes such as *Il1a*, *Il1b*, and *Il12b* downstream of TNF signaling, which may also contribute to protection [43,47–49].

Taken as a whole, our study deepens our understanding of the mechanisms by which TNF is able to position cells to better control intracellular bacterial infection. We find that cells which have been licensed by TNF rapidly undergo pyroptosis and robustly respond to flagellin-deficient *L. pneumophila* infection. We further characterize this death as being mediated not only by caspase-11, but additionally involving caspases-1 and -8, which together contribute to control of *L. pneumophila* replication in TNF-primed macrophages. We finally demonstrate that caspase-8 is required for bacterial control in a mouse model of pulmonary *L. pneumophila*

infection. Altogether, our findings highlight the multiple mechanisms by which TNF triggers protective death in cells through the activation of multiple caspases, and provide new insight into the function of TNF in host defense against intracellular *L. pneumophila* infection.

## Materials and methods

### Ethics statement

All animal experiments were carried out in accordance with the Federal regulations set forth in the Animal Welfare Act (AWA), recommendations in the NIH Guide for the Care and Use of Laboratory Animals, and the guidelines of the University of Pennsylvania Institutional Animal Use and Care Committee. All protocols used in this study were approved by the IACUC at the University of Pennsylvania (Protocol #804928, Protocol #804523).

### Bacterial culture

*Legionella pneumophila* Philadelphia 1 strains derived from the JR32 background or the LP02 *thyA* background [87] were cultured on charcoal yeast extract (CYE) agar containing streptomycin, as well as thymidine for the LP02 background strains, at 37˚C for 48 hours prior to infection. Wild-type strains as well as flagellin-deficient Δ*flaA* and Dot/Icm type IV secretion system-deficient Δ*dotA* mutant strains were used on both JR32 and LP02 genetic backgrounds [33,88].

### Mice

C57BL/6 mice were purchased from Jackson Laboratories. *Tnf*[-/-] mice, *Tnfr1*[-/-] mice, *Ripk3*[-/-] mice [43], *Ripk3*[-/-]*Casp8*[-/-] [43,54], *Ripk3*[-/-]*Casp8*[DA] [71], *Gsdmd*[-/-] [89], and *Gsdme*[-/-] [82] mice have been previously described and maintained as breeding lines in-house. *Gsdme*[-/-] mice were crossbred with *Gsdmd*[-/-] mice to produce *Gsdmd*[-/-]*Gsdme*[-/-] mice. *Ripk3*[-/-]*Casp8*[-/-] *Casp1*[-/-]*Casp11*[-/-] bone marrow was previously described [70] and graciously provided by Dr. Russell Vance. All animals were housed and bred in specific-pathogen-free conditions in accordance with the Animal Welfare Act (AWA) and the guidelines of the University of Pennsylvania Institutional Animal Use and Care Committee.

### Mouse infection

Mice between the ages of 8 and 12 weeks were anesthetized via intraperitoneal injection of 100 mg/kg ketamine and 10 mg/kg xylazine in PBS. Following confirmation of anesthetization, mice were infected through the intranasal route with 40 μL PBS carrying $1x10^6$ JR32 Δ*flaA* bacteria. Quantification of bacterial growth following infection was conducted by excision, weighing, and homogenization of lung tissue at the indicated timepoints using gentleMACS tissue dissociator (Miltenyi Biotec). CFUs were enumerated via plating of lung homogenates on CYE agar containing streptomycin.

### Mouse bone marrow-derived macrophage culture

Bone marrow was harvested from femurs, tibiae, and pelvises of mice described above. Bone marrow was suspended at $1x10^7$ cells/mL in 90% FBS, 10% DMSO solution for freezing in liquid nitrogen storage. Bone marrow cells were thawed and differentiated into macrophages by culture at 37˚C in media comprising RPMI, 30% L929 cell supernatant, 20% FBS, 100 IU/mL penicillin, and 100 μg/mL streptomycin. One day prior to infection, cells were plated in media comprising RPMI, 15% L929 cell supernatant, and 10% FBS. Macrophages were plated at $2x10^5$ cells per well in 24-well plates, $5x10^4$ cells per well in 48-well plates, or $1x10^5$ cells per

well in 96-well plates. Recombinant murine TNF-primed wells were plated in media containing 10 ng/mL rTNF for 16 hours prior to infection.

## Bacterial growth curves

For experiments analyzing bacterial growth restriction, infection of BMDMs with JR32 *ΔflaA* bacteria was carried out in 24-well plates at MOI = 1 in 500 μL macrophage plating media. At 1 hour following infection, cells were washed with warm RPMI to remove extracellular bacteria. Macrophages were lysed with sterile $diH_2O$ and lysates were serially diluted, then plated on CYE agar plates containing streptomycin. Bacterial CFUs were quantified following 4–5 days of incubation at 37°C and normalized relative to CFUs isolated at 1 hour post infection.

## Cell death assays

To measure cytotoxicity by way of lactate dehydrogenase (LDH) release, BMDMs were infected with LP02 *ΔflaA* bacteria in 48-well tissue culture plates. Release of LDH into the culture supernatant was quantified after infection using an LDH Cytotoxicity Detection Kit (Clontech). LDH release was normalized to mock-infected cells and cells treated with 1% Triton to establish maximum LDH release. To measure cytotoxicity by uptake of propidium iodide (PI), BMDMs were infected in 96-well black-walled tissue culture plates. At the time of infection, 5 μM PI was added to plate reader media (20 mM HEPES buffer and 10% FBS in Hank's Balanced Salt Solution). Cells were then allowed to equilibrate to 37°C for 10 minutes before being spun to the bottom of the plate at 1200 rpm for 5 minutes. PI uptake into cells was then measured at an excitation wavelength of 530 nm and an emission wavelength of 617 nm. PI uptake was normalized to mock-infected cells and 1% Triton-treated cells.

## Immunoblotting

To analyze protein expression and processing, cells were lysed directly with 1x SDS/PAGE sample buffer. Secreted proteins were isolated from cell supernatants by centrifugation at 2000 rpm for 10 minutes to remove cellular debris, followed by precipitation using trichloroacetic acid (TCA) overnight. Precipitated protein was pelleted by spinning at 13,000 rpm for 15 minutes at 4°C, then washed with ice-cold acetone, centrifuged at 13,000 rpm again for 10 minutes, before finally being suspended in 1x SDS/PAGE sample buffer. Samples were heated at 100°C for 5 minutes and then separated by SDS/PAGE and transferred to PVDF membranes (Millipore). Membranes were then probed with primary antibodies specific for murine caspase-11 (#C1354; Sigma-Aldrich), caspase-3 (#9662; Cell Signaling), caspase-8 (#4798; Cell Signaling), gasdermin D (#G7422; Sigma-Aldrich), IL-1β (12242S; Cell Signaling), and β-actin (#4967; Cell Signaling). Membranes were then probed with secondary antibodies anti-rat IgG (7077S; Cell Signaling), anti-mouse IgG (7076S; Cell Signaling), or anti-rabbit IgG (7074S; Cell Signaling). ECL Western Blotting Substrate and SuperSignal West Femto Substrate (Thermo Scientific) were used.

## ELISAS

Harvested cell supernatants were assayed using ELISA kits for murine IL-1α, IL-1β, and IL-6 (BD Biosciences).

## Caspase activity assay

Activity of caspases-3/7 and -8 were assessed using the corresponding Caspase-Glo Assays in white-walled 96-well plates (Promega).

### Statistical analysis

Graphing and statistical analysis were carried out in GraphPad Prism 7.0. In comparisons between two groups, unpaired Student's t-test was utilized to determine significance. In comparisons between more than two groups, two-way ANOVA was utilized to determine significance, with Tukey HSD test following up. Difference considered significant when the *P* value is $< 0.05$.

### Supporting information

**S1 Fig. TNFR1, but not NADPH oxidase or iNOS, is required to restrict intracellular *L. pneumophila* replication. (A)** WT, *Tnfr1⁻/⁻*, and *Tnfr1⁻/⁻Tnfr2⁻/⁻* BMDMs were infected with replicating Δ*flaA L. pneumophila* at MOI = 1. The fold change in CFUs was quantified at 72 hours post-infection. **(B-D)** WT, *Nos2⁻/⁻*, and *Cybb⁻/⁻* BMDMs were infected with replicating (B, D) or non-replicating (C) Δ*flaA L. pneumophila* at MOI = 1 (B,D) or 10 (C). The fold change in CFUs was quantified at 72 hours post-infection, while TNF secretion was assessed at 16 hours post-infection. * is p<0.05 ** is p<0.01, and *** is p<0.001 by one-way ANOVA with Tukey HSD post-test (A,B) or Student's t-test (D), ns is not significant.
(TIF)

**S2 Fig. The *L. pneumophila* type-IV secretion system is required for induction of BMDM cell death.** WT BMDMs were uninfected or infected with non-replicating *L. pneumophila* Δ*flaA* or Δ*dotA* mutant strains at MOI = 10 for 16 hours. Cells were primed with either 10 ng/mL rTNF or PBS mock control for 16 hours prior to infection.
(TIF)

**S3 Fig. Inflammasome-related gene expression following TNF priming and *L. pneumophila* infection.** WT cells were primed with either 10 ng/mL rTNF or PBS for 16 hours, then infected with non-replicating Δ*flaA L. pneumophila* at MOI = 50. Supernatants and cell lysates were collected for immunoblot and RT-qPCR analysis.
(TIF)

**S4 Fig. GBPs on chromosome 3 are dispensable for TNF-licensed cell death and bacterial restriction.** WT and GBP^chr3 BMDMs were primed with either 10 ng/mL rTNF or PBS mock control for 16 hours prior to infection with non-replicating (A) or replicating (B) Δ*flaA L. pneumophila* at MOI = 10 (A) or 1 (B). **(A)** Cytotoxicity was measured by LDH release assay. **(B)** The fold change in CFUs was determined at 48 hours post-infection. Data shown are representative of two independent experiments.
(TIF)

**S5 Fig. Caspase-8 contributes to TNF-licensed release of IL-1 family cytokines, but not expression of caspase-11, following *L. pneumophila* infection.** *Ripk3⁻/⁻* and *Ripk3⁻/⁻Casp8⁻/⁻* BMDMs were primed with either 10 ng/mL rTNF or PBS mock control for 16 hours prior to infection with non-replicating Δ*flaA L. pneumophila* at MOI = 50. **(A)** Supernatants were analyzed by ELISA at 24 hpi. **(B)** BMDMs were lysed and analyzed by immunoblot for caspase-11 protein at indicated timepoints. Graphs show the mean ± SEM of triplicate wells. * is p<0.05, ** is p<0.01, and *** is p<0.001 by 2-way ANOVA with Tukey HSD post-test.
(TIF)

**S6 Fig. Caspase-8 is required for maximal cell death following *L. pneumophila* infection.** *Ripk3⁻/⁻*, *Ripk3⁻/⁻Casp8⁻/⁻*, and *Ripk3⁻/⁻Casp8^DA* cells were infected with non-replicating *L. pneumophila* Δ*flaA* mutant strain at MOI = 10 for 8 hours. Cells were primed with either 10 ng/mL

rTNF or PBS mock control for 16 hours prior to infection.
(TIF)

## Acknowledgments

We would like to acknowledge the labs of Sunny Shin and Igor Brodsky for frequent scientific insight and sharing of reagents. We also thank James Grayczyk for *Gsdmd⁻ᐟ⁻Gsdme⁻ᐟ⁻* mice. We thank Jessica Doerner for help with pulmonary infection. We thank members of the Brodsky lab for mouse bone marrow, as well as Russell Vance for *Ripk3⁻ᐟ⁻Casp1⁻ᐟ⁻Casp11⁻ᐟ⁻* and *Ripk3⁻ᐟ⁻Casp1⁻ᐟ⁻Casp11⁻ᐟ⁻Casp8⁻ᐟ⁻* triple and quadruple knockout bone marrow.

## Author Contributions

**Conceptualization:** Tzvi Y. Pollock.

**Data curation:** Tzvi Y. Pollock.

**Formal analysis:** Tzvi Y. Pollock, Víctor R. Vázquez Marrero.

**Funding acquisition:** Tzvi Y. Pollock, Sunny Shin.

**Investigation:** Tzvi Y. Pollock, Víctor R. Vázquez Marrero.

**Methodology:** Tzvi Y. Pollock.

**Project administration:** Sunny Shin.

**Resources:** Igor E. Brodsky, Sunny Shin.

**Supervision:** Igor E. Brodsky, Sunny Shin.

**Validation:** Tzvi Y. Pollock, Víctor R. Vázquez Marrero.

**Visualization:** Tzvi Y. Pollock.

**Writing – original draft:** Tzvi Y. Pollock.

**Writing – review & editing:** Tzvi Y. Pollock, Igor E. Brodsky, Sunny Shin.

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
