## [Decision Letter · Decision Letter 0]

11 Sep 2022

Dear Dr. Shin,

Thank you very much for submitting your manuscript "TNF licenses macrophages to undergo rapid caspase-1, -11, and -8-mediated cell death that restricts Legionella pneumophila infection" for consideration at PLOS Pathogens. As with all papers reviewed by the journal, your manuscript was reviewed by members of the editorial board and by several independent reviewers. In light of the reviews (below this email), we would like to invite the resubmission of a significantly-revised version that takes into account the reviewers' comments.

The reviewers appreciated the impact and rigor of your study and felt that it has strong potential to advance the field. Many of the comments relate to providing additional context or more carefully examining interpretation of the data. In addition to any text modifications, please be sure to address the major points below:

(1) clarification as to the contribution of ROS to host cell death

(2) additional assays to better identify the cell death pathway(s) involved

(3) Activation assays specific to caspase-11 as noted by reviewer #3.

We cannot make any decision about publication until we have seen the revised manuscript and your response to the reviewers' comments. Your revised manuscript may be sent to reviewers for further evaluation.

Sincerely,

Mary X. O'Riordan

Associate Editor

PLOS Pathogens

Raphael Valdivia

Section Editor

PLOS Pathogens

Kasturi Haldar

Editor-in-Chief

PLOS Pathogens

orcid.org/0000-0001-5065-158X

Michael Malim

Editor-in-Chief

PLOS Pathogens

orcid.org/0000-0002-7699-2064

The reviewers appreciated the impact and rigor of your study, and felt that it has strong potential to advance the field. Many of the comments relate to providing additional context or more carefully examining interpretation of the data. In addition to any text modifications, please be sure to address the major points below:

(1) clarification as to the contribution of ROS to host cell death

(2) additional assays to better identify the cell death pathway(s) involved

(3) Activation assays specific to caspase-11 as noted by reviewer #3.

Reviewer's Responses to Questions

**Part I - Summary**

Reviewer #1: In this study by Pollock et al., the authors address the role of TNF in mediating control of intracellular Legionella by pathways that lead to death of infected macrophages. The manuscript is very well-written and the data are convincing that at least one major mechanism by which TNF contributes to host defense against Legionella is licensing macrophages to undergo rapid caspase-1, -11, and -8-mediated macrophage death. In addition, they show that TNFR1 signaling is required for restriction of Legionella infection of macrophages and primes cells to rapidly undergo cell death. They go on to demonstrate that the T4SS is somehow triggering caspase-1 and -11 activation in TNF-primed macrophages and that TNF upregulated the expression of IL1a, IL1b, and IL18 mRNA as well as the expression of Caspase-11 protein. They also show that in macrophages from GbpChr3-/- mice that cell death and bacterial control were not abrogated. These findings suggest that TNF-dependent upregulation of inflammasome components leads to rapid engagement of non-canonical inflammasome activation independent of GBPs on chromosome 3. Finally, in a series of elegant experiments utilizing various BMDMs from KO and transgenic mice that caspase-8 contributes to cell death independently of its autocleavage downstream of TNF signaling. Finally, they demonstrate that TNR1 and caspse-8 are required for control of pulmonary Legionella infection in a mouse model. The manuscript is excellent and the findings are a significant contribution to the field and I have only minor suggestions to address before publication in Plos Pathogens.

Reviewer #2: In this manuscript, Pollock et al demonstrate a novel role for TNF in caspase-mediated cell death in response to Legionella pneumophila infection. It is clear that TNF is important for pulmonary control of L. pneumophila, and the data presented reveal that caspase-1, -11, and 8 are essential for NAIP5/NLRC4-independent TNF-mediated restriction. The manuscript is well-written and the data are rigorous. I have some comments on interpretation of the data and further discussion of these data in the context of the published literature, which will further support/refine the model. Overall, this work is an important contribution to the field of antimicrobial innate immunity and host response to intracellular bacterial pathogens.

Reviewer #3: This is a comprehensive study focused or the role of TNF licensing in mediating macrophage death in response to L. pneumophila. The authors demonstrated that TNF priming triggers cell death, however, the downstream mechanisms remain unclear. They show that TNF signaling licenses macrophages to die rapidly in response to Legionella infection mainly by rapid gasdermin-dependent, pyroptotic death downstream of inflammasome activation. They also show that TNF signaling upregulates components of the inflammasome response. They suggest that caspase-11 is rapidly activated, while caspase-1 and caspase-8 mediate delayed pyroptotic death. Furthermore, caspase-8 is required for control of pulmonary Legionella infection. Together, they propose that all three caspases are collectively required for optimal TNF-mediated restriction of L. pneumophila replication in macrophages. These findings reveal a TNF-dependent mechanism in bone marrow derived macrophages for the activation of rapid cell death that is collectively mediated by caspases-1, -8, and -11 and subsequent restriction of Legionella infection. While the findings are mechanistically important, the authors need to address some points.

**Part II – Major Issues: Key Experiments Required for Acceptance**

Reviewer #1: NONE

Reviewer #2: Lines 144-154: These data do not rule out a role for ROS since mtROS are not evaluated (and are upregulated downstream of TNF signaling). To support this conclusion, additional data are required (antioxidant treatment) or the text should be modified to specify NOX2-mediated ROS (and not total ROS). Is the quantity of TNF secreted consistent between wt and Nos2-/- and cybb-/- BMDMs? Do iNOS or NOX2 contribute to bacterial restriction in TNF-primed BMDMs? The authors should also consider discussing their work in the context of the study by Roca et al (PMID: 31474371) showing a role for mtROS in TNF-mediated mediated cell death and restriction of Mycobacterium spp.

Lines 306-310: It's difficult to compare cell death between Ripk3-/-Casp8DA, Ripk3-/-, and Ripk3-/-Casp8-/- since they are plotted on separate histograms. Plotting all of the PI-uptake data on the same histogram would be a mess, so showing LDH data here would be more intuitive for comparison between these BMDMs. The authors could consider including the PI uptake data as supplemental information.

Lines 312-315/Fig 4D: Could decreased Casp3/7 activity in primed cells be a consequence of less viable cells? I'm not sure how well this assay detects enzymatic activity in cell supernatants. Can the authors comment on this? Perhaps it would be worth normalizing these data to viable cells?

Lines 344-345: It seems that gasdermin-dependent cell death is not responsible for L. pneumophila restriction. How do the authors reconcile this with their overall model whereby TNF-mediated cell death is responsible for bacterial restriction? It would seem that the 'additional TNF-mediated protective mechanisms' are quite important here. In Fig 5B, it seems that there is some gasdermin-independent cell death at 8h post-infection. I suggest modifying lines 357-359 to address this. Based on the work of Oxenius and colleagues, it may be phagolysosomal fusion (PMID: 27105352). It would be interesting to see whether this is the case, but I recognize it is likely out of the scope of the study. Please provide additional discussion on potential alternative mechanisms of bacterial restriction.

Reviewer #3: - The authors did not examine other specific forms of cell death including PANoptosis and the downstream repercussions. LDH release and PI uptake can occur in response to several forms of cell death.

- The authors mention in the legends of several figures using non-replicating fla mutants. They did not explain the characteristic of this unusual fla mutant. Typically, Legionella fla mutants replicate very well even in restrictive cells.

- Please add the name of the different mutants above every panel. It is confusing to follow which ones included WT legionella, DotA mutant or Fla mutant.

- In several figures, they focus on IL-1a and not IL-1B. Although IL-1B is the well characterized inflammasome-dependent cytokine whereas IL-1a is not.

- The authors conclude that TNF licenses macrophages to undergo caspase-1 and 11-mediated cell death to restrict Legionella, yet in figure 2 E, primed Casp-1-/-/casp11-/- macrophages still restrict Legionella replication.

- The authors consider the increase of caspase-11 expression as a sign of activation, which is incorrect. They also use the cleaved caspase-11 band on western blots as a sign of activation, which is also incorrect, since it is known in the field that cleavage of caspase-11 does not signify activation. Instead, they need to use activation assays specific to caspase-11 as published by others.

- It is not clear why caspase-11 cleavage products are detected in culture supernatants and not in cell lysates (Fig. 3G).

- There is a difference between the secretion of cytokines and the passive release of cytokines from dying cells. The authors need to accurately state these differences and use them appropriately through the text while explaining their results.

- SFig 4B is confusing. Why is there no actin in the cell lysates? What is the band that they show as casp11? Why does it disappear when actin disappears?

- The authors need to explain how can “uncleavable” caspase-8 modulate Legionella replication?

**Part III – Minor Issues: Editorial and Data Presentation Modifications**

Reviewer #1: 1. The authors show that triggering cell death pathways in macrophages is dependent on the T4SS. Can they speculate what the host might be sensing that then leads to activation of caspases-1 and -11 and finally caspase-8? What is upstream that is dependent on TNF, but independent of GBPs on chrom3?

2. In Figure 6, is it possible to look in infected lungs and quantitate macrophage death to see differences in WT compared to KO mice? And does this correlate with differences in influx of cells that can kill Legionella e.g., perhaps higher levels of neutrophils in WT compared to KO’s?

Reviewer #2: Lines 180-182: It seems that only PI uptake assays are shown. Please also include the LDH-based cytotoxicity data.

Lines 202-205: The statement '… other caspase-1/11-independent factors contribute to TNF-mediated cell death…' is confusing in light of the data in Fig 2B-C and the statement in lines 188-190. Can the authors please clarify what the statement is based on?

Lines 251-263: Is there evidence that these GBPs are actually present in infected BMDMs in the absence of IFN-g priming? Fig S3B: Why are fold CFU at 48 h shown as opposed to 72 h like the other figures?

Figure 4D: It would be helpful to make the Tnf-/- primed bars a different color since it looks a lot like the red used for Tnf-/- mock.

Figure 5B. Please comment on gasdermin-independent cell death observed at 8h post-infection in TNF-primed cells

Lines 343-345: The data in Fig 5D suggest to me that bacterial restriction is independent of cell death

Lines 460-462: What do the authors think the pro-inflammatory genes could be?

Line 509: What was the MOI used for Fig 1H?

Reviewer #3: - The sentence: “Using this priming and infection model alongside genetic tools, our data demonstrate that TNF signaling through TNFR1 licenses L. pneumophila-infected cells to rapidly and robustly undergo flagellin and NAIP/NLRC4-independent cell death in a T4SS-dependent manner” Is somewhat misleading, since it is known that T4SS is indispensable for Legionella replication under all circumstances.

- “TNF is able to license this pyroptotic death in part by upregulating components of the caspase-11 non-canonical inflammasome ahead of infection” is inaccurate since the authors did not exclude other types of cell death such as necrosis and apoptosis (PANoptosis).

- Please make sure to add the appropriate references for the role of flagellin in Legionella replication and its recognition by Nlrc4/Ipaf.

- Please discuss the findings in relation to other similar studies in the field such as Mascarenhas et al 2017, Santic ela al 2007, kawamoto et al 2017,

PLOS authors have the option to publish the peer review history of their article (what does this mean?). If published, this will include your full peer review and any attached files.

Reviewer #1: No

Reviewer #2: No

Reviewer #3: No
---

## [Decision Letter · Decision Letter 1]

25 May 2023

Dear Dr. Shin,

We are pleased to inform you that your manuscript 'TNF licenses macrophages to undergo rapid caspase-1, -11, and -8-mediated cell death that restricts *Legionella pneumophila* infection' has been provisionally accepted for publication in PLOS Pathogens.

Best regards,

Raphael H. Valdivia

Section Editor

PLOS Pathogens

Raphael Valdivia

Section Editor

PLOS Pathogens

Kasturi Haldar

Editor-in-Chief

PLOS Pathogens

orcid.org/0000-0001-5065-158X

Michael Malim

Editor-in-Chief

PLOS Pathogens

orcid.org/0000-0002-7699-2064

Reviewer Comments (if any, and for reference):

Reviewer's Responses to Questions

**Part I - Summary**

Reviewer #1: The authors have done an excellent job of addressing the reviewer's concerns.

Reviewer #2: The authors have addressed all of my concerns and I congratulate them on a rigorous and impactful study.

Reviewer #3: the authors responded to all my comments.

**Part II – Major Issues: Key Experiments Required for Acceptance**

Reviewer #1: (No Response)

Reviewer #2: (No Response)

Reviewer #3: none

**Part III – Minor Issues: Editorial and Data Presentation Modifications**

Reviewer #1: (No Response)

Reviewer #2: (No Response)

Reviewer #3: none

PLOS authors have the option to publish the peer review history of their article (what does this mean?). If published, this will include your full peer review and any attached files.

Reviewer #1: No

Reviewer #2: No

Reviewer #3: No

---

## [Editor Report · Acceptance letter]

4 Jun 2023

Dear Dr. Shin,

We are delighted to inform you that your manuscript, "TNF licenses macrophages to undergo rapid caspase-1, -11, and -8-mediated cell death that restricts *Legionella pneumophila* infection," has been formally accepted for publication in PLOS Pathogens.

Best regards,

Kasturi Haldar

Editor-in-Chief

PLOS Pathogens

orcid.org/0000-0001-5065-158X

Michael Malim

Editor-in-Chief

PLOS Pathogens

orcid.org/0000-0002-7699-2064